# Case Reports of Aortic Aneurism in Fragile X Syndrome

**DOI:** 10.3390/genes13091560

**Published:** 2022-08-30

**Authors:** Sivan Lewis, Andrew DePass, Randi J. Hagerman, Reymundo Lozano

**Affiliations:** 1Department of Genetics and Genomic Sciences, Icahn School of Medicine at Mount Sinai, New York, NY 10029, USA; 2MIND Institute and Department of Pediatrics, University of California Davis Health, Sacramento, CA 95817, USA; 3Department of Pediatrics, Icahn School of Medicine at Mount Sinai, New York, NY 10029, USA

**Keywords:** aortic aneurysm, aortic dilatation, fragile X syndrome, intellectual disability, connective tissue disorder

## Abstract

Fragile X syndrome (FXS) is an inherited genetic condition that is the leading known cause of inherited intellectual developmental disability. Phenotypically, individuals with FXS also present with distinct physical features including, elongated face, prominent ears, pectus excavatum, macroorchidism, and joint laxity, which suggests connective tissue dysplasia. In addition to mitral valve prolapse, aortic dilatation has been identified within individuals with FXS. Abnormal elastin fiber networks have been found in the skin, valves, and aorta in individual cases. Aortic dilatation has been described in other connective tissue disorders, particularly Marfan syndrome. However, while aortic aneurysms are characteristic of Marfan syndrome, no similar cases have been reported in FXS patients to date. This case report details the presentation of two patients with FXS and aortic aneurysm. Our two cases highlight the risks of aortic pathology in FXS, and the need for monitoring in asymptomatic patients with significant aortic dilatation.

## 1. Introduction

Fragile X syndrome (FXS) is the leading cause of inherited intellectual developmental disability (IDD) and autism spectrum disorder (ASD) [1,2]. FXS is an X-linked inherited condition [3]. FXS presents genetically through the emergence of CGG repeats in the 5′ untranslated region of the fragile X messenger ribonucleoprotein 1 (*FMR1*) gene located on chromosome Xq27.3 [3]. The CGG repeat size is variable between people, and usually healthy people have less than 45 repeats [4]. Affected individuals possess 200 CGG repeats or greater [5]. Repeats in conjunction with the methylation mediated silencing of the *FMR1* promoter decreases expression [6], and culminates in a net reduction in *FMR1* protein (FMRP) [7].

Phenotypically, individuals with FXS present with distinct physical features, including elongated face, prominent ears, pectus excavatum, macroorchidism, and joint laxity, which can be indicative of connective tissue dysplasia [8]. There is also increased incidence of mitral valve prolapse (MVP) and aortic root dilatation [9].

Puzzo et. al. studied a cohort of 13 individuals with FXS between early childhood and adolescence and MVP was identified in 77% of those affected [10]. In another study by Loehr et. al., examining a larger cohort of 40 individuals with FXS, 13–55% of patients had mitral valve prolapse with the frequency increasing with age, and MVP was present in 80% of patients over the age of 18 years [11]. In addition to MVP, aortic dilatation has also been identified within individuals with FXS. Abnormal elastin fiber networks have been found in the skin, valves, and aorta in individual cases [9,12,13,14]. Aortic dilatation is defined as dimension of the aortic diameter that is greater than the 95th percentile for the normal person’s age, sex, and body size [15]; it is an age-dependent phenomenon that leads to variation in cohorts. Aortic dilatation has been identified in the ascending aorta in two patients aged 17 and 23 years [9], and in the aortic root of 52% of patients within a 23-patient cohort of ages between 18–80 years, with a mean age of 51 years [10].

These cardiac abnormalities have been described in other connective tissue disorders, particularly Marfan syndrome [16]. However, while aortic aneurysms, defined as a localized dilatation of the aorta greater than 1.5 times the expected diameter [15] are characteristic of Marfan syndrome [17], no similar cases have been reported in FXS patients. This case report details the presentation of two patients with FXS and aortic aneurysms seen at the Mount Sinai Hospital and the University of Colorado Hospital.

## 2. Case Presentation

### 2.1. Patient #1

This is the case of a 43-year-old man, with fragile X full mutation with 350 CGG repeats. He presented with longstanding problems of anxiety, intellectual disability, and a significant recent history of aortic aneurysm diagnosed at the age of 42 years.

The patient was diagnosed with FXS at the age of 31 years after his mother tested positive as a fragile X premutation carrier. His mother never had symptoms of premature ovarian insufficiency, and no movement-related concerns were reported. However, she reported chronic pain, and chronic fatigue. Additional problems included polycythemia vera.

Retrospectively, patient 1 reported panic attacks in childhood with any change in routine, and this persisted into adulthood. He has seen a therapist, but not for the past year. He experienced increased anxiety since the COVID pandemic started, and was started on a low dose of sertraline, with slight improvement. Cognitive issues have also been reported; the patient is unable to travel by himself or handle his own finances. His medical history was also notable for strabismus, reflux, asthma, hypertension, and polycythemia. Developmental history was remarkable for speech and motor delay, with services including speech therapy, physical therapy, and occupational therapy provided from 2 years through to 21 years of age. The patient also endorsed a recent weight gain of 8 kgs in the past 9 months, with a BMI increase from 28 to 31.

At the age of 42 years, 1 year prior to establishing care at our clinic, his primary care physician noted a new murmur. The patient underwent an echocardiogram and computed tomography of the heart, which showed a bicuspid aortic valve, a 5.4 cm aortic root, and an ascending aorta aneurysm. The patient subsequently underwent aortic valve replacement, aortic root replacement, and ascending aorta replacement with hemiarch replacement. The histologic examination showed medial degeneration, with myxoid material accumulation at the ascending aorta and mucoid degeneration of the aortic valve. The recovery has been uneventful, and the patient continued with cardiac rehabilitation at the time of consultation with us. Genetic work up sent by his cardiologist found a heterozygous variant of uncertain significance in NOTCH1, namely c.G6377C (p.G2126A).

### 2.2. Patient #2

This patient was diagnosed with FXS in childhood and had the clinical phenotype including large and prominent ears with cupping of the ear pinnae, hyperextensible finger joints, and flat feet. He had special education support and therapies throughout his schooling.

He died at age 40, but was living independently for 20 years working at a thrift store within walking distance from his condominium. He was never late for work. He was high functioning. He loved books and was able to read independently. He had regular, yearly checkups with his primary care doctor, and no problems were diagnosed.

Three days before his death he complained of chest pain and had diaphoresis, an episode of vomiting and syncope. His chest pain was worse when lying down. He was seen in the local emergency room with a blood pressure of 130/107, heart rate of 73, height of 1.753 m, and weight of 104.32 kg. Imaging demonstrated a large aortic aneurysm and he was transferred to a large metropolitan hospital. There, CT imaging demonstrated a Stanford type A ascending thoracic aortic dissection with a dilated aortic root measuring 6.9 cm. He also had a poorly characterized flap anteriorly situated at the aortic root, extending toward the right coronary sinus, and a 2nd discontinuous dissection flap within the posterior aortic arch extending into the left brachiocephalic artery. There was also a moderate-sized pericardial effusion with a mean Hounsfield value of approximately 25. The pericardial effusion measured up to 1.9 cm in thickness, presumably representing a hemopericardium. His PT and PTT were normal but his D-dimer quantitative level was 1100 (the normal range is 0–500 ng/mL). He was immediately taken to surgery, where a Bentall procedure was carried out successfully; however, he died 2 days after surgery.

## 3. Discussion

The findings of mitral valve prolapse and aortic root dilatation in a significant percentage of patients with FXS supports the hypothesis of a connective tissue dysfunction in this condition. Our two cases of ascending aorta dilatation also support this hypothesis.

Elastin abnormalities have been observed in the skin of patients with FXS [13], and in a case report of an 18-year-old male with FXS who died of cardiac arrest [14].

However, little is known about the mechanism by which a decrease in FMRP expression causes connective tissue dysregulation.

Malecki et al. [18] suggested that pathways and proteins that contribute to elastin fragmentation in Marfan syndrome can also explain the relationship between FMRP and connective tissue dysregulation. MMPs (matrix metalloproteinases) are proteinases involved in fragmentation of elastic fibers in Marfan syndrome, with MMP-2 and MMP-9 specifically involved in the pathogenesis of Marfan aneurysm formation. In aneurysmal aortic tissue from Fbn1-deficient mice, Chung et al. [19] found that upregulation of MMP-2 and MMP-9 was accompanied by severe elastic fiber fragmentation and degradation of aortic contraction, possibly explaining the pathogenesis of aortic aneurysm.

MMP-9 mRNA has also been identified as an FMRP target [20,21], and the presence of MMP-9 mRNA in a complex with FMRP has been demonstrated in murine hippocampal regions. *Fmr1* KO mice, as well as human FXS post-mortem brain samples, have shown increased MMP-9 protein expression compared with controls [22,23]. In *Fmr1/MMP-9* double KO mice both neural as well as non-neural abnormalities associated with FXS were corrected, suggesting that aberrant regulation of MMP-9 expression may also contribute to the non-neural features of FXS [23].

Further investigation into the role of FMRP in regulating MMP-9 translation in vasculature and cardiac valves is warranted, especially given the difference in natural history of aortic dilatation and aneurysms between Marfan and FXS.

In patients with Marfan syndrome, aortic root dilatation leading to aneurysm formation and fatal dissection is not rare [17]. However, aortic aneurysms have not been reported in FXS despite of the presence of cardiac lesions similar to those in Marfan syndrome [16].

The average life expectancy of an individual with Marfan syndrome is approximately 50% of normal life expectancy, with cardiovascular problems as the leading cause of death [17]. Whether cardiac abnormalities significantly alter life expectancy in FXS patients is yet to be determined. Waldstein and Hagerman [14] reported a case of an 18-year-old male with FXS and cardiomegaly, abnormal mitral valve, tubular hypoplasia, and coarctation of the aorta who died of viral myocarditis. Sabaratnam [24] reported two patients with mitral valve abnormalities in which mortality occurred at the ages of 67 years from myocardial ischemia and 87 years from pulmonary embolism, suggesting a lesser degree of life span shortening compared to Marfan syndrome.

Our two cases highlight the risks of aortic pathology in FXS, and the need for monitoring asymptomatic patients with significant aortic dilatation [Table 1].

Patient 1 was incidentally found to have an aortic aneurysm. His risk factors included male gender and, more importantly, hypertension, yet he was a non-smoker and presented at a relatively young age [25]. Genetic evaluation revealed a heterozygous variant of uncertain significance in *NOTCH1*, namely c.G6377C (p.G2126A). It is important to note that our patient also had a bicuspid aortic valve, and mutations in *NOTCH1* have been associated with isolated and familial bicuspid aortic valve and aortic aneurysm [26]. However, the c.G6377C variant of uncertain significance identified in the *NOTCH1* gene in Case 1 has not been reported as pathogenic in the literature, and was rare in large healthy population databases. In silico analysis also supported that this missense variant does not damage protein function. It is unknown whether *NOTCH1* variants affect patients with FMRP deficiency differently, as aberrant expression of NOTCH1 has been demonstrated in fragile X human embryonic stem cells [27]. Moreover, while *NOTCH1* has does not have a defined role in pathogenesis of Marfan syndrome, *NOTCH1* haploinsufficiency has been shown to exacerbate the aortic root dilatation in the Marfan syndrome mouse model [28]. Based on the current information in the literature, the variant is not classified as pathogenic, and the most likely genetic etiology of aortic aneurysm in this patient remains FXS.

Patient 2 died of an aortic aneurysm and aortic dissection, and although he was not found to be hypertensive before his chest pain, he had a BP of 130/107 in the ER. He was also obese. As the patient was high functioning, he possibly had mosaicism with some FMRP production and elevated mRNA; however, this was not tested. There have been three cases of female premutation carriers who presented with sudden severe chest pain and were diagnosed with spontaneous coronary artery dissection (SCAD) [29]; thus, dissections may also occur in carriers.

It is known that elevated levels of MMP-9 in FXS can be lowered by either minocycline or metformin, and both are targeted treatments for FXS. Future research is necessary to determine whether treatment with one of these medications should be considered in patients with significant aortic arch dilatation.

## Figures and Tables

**Table 1 genes-13-01560-t001:** Patient Characteristics.

Characteristic	Patient 1	Patient 2
Age	43	40
Mass (kg)	96.7	104.32
Height (cm)	175.3	175.3
BMI	31.8	33.9
Hypertension	Treated	BP 130/107 upon presentation with dissection
Aortic aneurysm Diameter (cm)	5.4	6.9
Valvular abnormalities	Bicuspid aortic valve	
CGG Repeats	350	
Other genetic test results	*NOTCH1* p.G2126A	
Medical history	Intellectual disability, polycythemia, strabismus, asthma, GERD, anxiety	Mild intellectual disability

Controlled blood pressure is defined here as blood pressure < 140/90 mmHg; normal = BMI 18.5–24.9, overweight = BMI 25.0–29.9, obese = BMI > 30.0; BMI = body mass index, BP = blood pressure.

## Data Availability

Not applicable.

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
