# Peer review of "Case Reports of Aortic Aneurism in Fragile X Syndrome"

_genes, 2022, doi:10.3390/genes13091560_

Round 1

Reviewer 1 Report

This study detailed the presentation of two patients with FXS and aortic aneurysm. These two cases highlighted the risks of aortic pathology in FXS and the need for monitoring in asymptomatic patients who have significant aortic dilatation.

Strengths:
This study provided a new point about the pathophysiology of FXS, and the authors tried to discuss the correlation of genes in FXS with the connective tissue dysplasia, which is interesting. The case reports of these two patients are clearly and detailed.

Weakness:

1.      In the abstract and introduction, the authors tried to talk about the similarity and differences between Marfan Syndrome and FXS. In the discussion, they mentioned the genes related to FXS, including MMP-9 and NOTCH1, and they think these genes may be related to the aortic aneurysms observed in these patients with FXS. If there are similar phenotypes in Marfan Syndrome and FXS, are MMP-9 and NOTCH1 also involved in Marfan Syndrome? Could the authors add some information from literatures about this question?

2. The information about confidentiality of processing the case information should be provided in the manuscript.

Author Response

In the abstract and introduction, the authors tried to talk about the similarity and differences between Marfan Syndrome and FXS. In the discussion, they mentioned the genes related to FXS, including MMP-9 and NOTCH1, and they think these genes may be related to the aortic aneurysms observed in these patients with FXS. If there are similar phenotypes in Marfan Syndrome and FXS, are MMP-9 and NOTCH1 also involved in Marfan Syndrome? Could the authors add some information from literatures about this question? 

ADDED - please see 2 new references and comments.

Moreover, while NOTCH1 has does not have a defined role in pathogenesis of Marfan syndrome, NOTCH1 haploinsufficiency has been shown to exacerbate the aortic root dilatation in the Marfan syndrome mouse model [28].

  1. The information about confidentiality of processing the case information should be provided in the manuscript. 

Confidentiality Statement was added.

Confidentiality Statement: All records/data were kept confidential in accordance with HIPPA law. Identifiable health information was stored securely on institutional electronic medical record with password protected access enabled.

Reviewer 2 Report

line 2: change to "Case Reports of Aortic Aneurysm in Fragile X Syndrome" 

In general, all case reports are accompanied by a literature review that backs up your case report findings. It is not necessary to indicate "review of the literature" in your title. Otherwise, you will need to indicate your methods of literature review within your article to increase the strength of your article.

line 15/16/38/43/45/48/53/132/174/keywords vs line 21/133/146: in some areas you use "dilation", in some others you use "dilatation". You need to use consistent terminology throughout, rather than use these two terms interchangeably. In general, "dilatation" is the correct usage, as "dilation" is used more for normal physiology, whereas dilatation is more referring to pathologic processes. 

line 33 - remove "induced", change "decrease" to "decreases"

line 62 - use FXS instead of "Fragile X", as you already defined the abbreviation

line 70 - sertraline is spelled incorrectly

line 80 and 81 - "aortic" does not need to be capitalized

line 81 - "hemiarch" does not need to be capitalized

line 84 - change "rehab" to "rehabilitation" -- this is a formal paper

line 93 - change "condo" to "condominium" -- this is a formal paper

line 93/94 - run on sentence. Change to "He was high functioning. He loved books and was able to read independently."

line 100 - need a comma after "There"

line 104 - brachiocephalic is spelled incorrectly

line 106/107 - hemo-pericardium spelled incorrectly

line 113 - add "Our two cases of ascending aorta dilatation also help to support this hypothesis."

line 118 - do not capitalize "al"

line 138 - use "FXS" instead of "fragile x"

line 139 - change "yo" to "year old"

line 151 - bicuspid is spelled incorrectly

line 156 - "affect", not "affects"

line 159 - change "and" to "but is"

line 165-167 - run on sentence. Need to rewrite.

All your references are numbered twice.

Author Response

Thank you so much for all the comments. All the grammatical revisions were accepted and marked in the manuscript with track changes. The terminology is now also consistent with "dilatation".

line 15/16/38/43/45/48/53/132/174/keywords vs line 21/133/146: in some areas you use "dilation", in some others you use "dilatation". You need to use consistent terminology throughout, rather than use these two terms interchangeably. In general, "dilatation" is the correct usage, as "dilation" is used more for normal physiology, whereas dilatation is more referring to pathologic processes. 

line 33 - remove "induced", change "decrease" to "decreases"- Accepted

line 62 - use FXS instead of "Fragile X", as you already defined the abbreviation - Accepted

line 70 - sertraline is spelled incorrectly- Accepted

line 80 and 81 - "aortic" does not need to be capitalized- Accepted

line 81 - "hemiarch" does not need to be capitalized- Accepted

line 84 - change "rehab" to "rehabilitation" -- this is a formal paper- Accepted

line 93 - change "condo" to "condominium" -- this is a formal paper- Accepted

line 93/94 - run on sentence. Change to "He was high functioning. He loved books and was able to read independently."- Accepted

line 100 - need a comma after "There"- Accepted

line 104 - brachiocephalic is spelled incorrectly- Accepted

line 106/107 - hemo-pericardium spelled incorrectly- Accepted

line 113 - add "Our two cases of ascending aorta dilatation also help to support this hypothesis."- Accepted

line 118 - do not capitalize "al"- Accepted

line 138 - use "FXS" instead of "fragile x"- Accepted

line 139 - change "yo" to "year old"- Accepted

line 151 - bicuspid is spelled incorrectly- Accepted

line 156 - "affect", not "affects"- Accepted

line 159 - change "and" to "but is", changed by "therefore"

line 165-167 - run on sentence. Need to rewrite. It was rewriten - Accepted